# Resistance Mechanism to Metsulfuron-Methyl in *Polypogon fugax*

**DOI:** 10.3390/plants10071309

**Published:** 2021-06-28

**Authors:** Xiaoyue Yu, Hanwen Wu, Jianping Zhang, Yongjie Yang, Wei Tang, Yongliang Lu

**Affiliations:** 1State Key Laboratory of Rice Biology, China National Rice Research Institute, Hangzhou 311400, China; yuxiaoyue@caas.cn (X.Y.); zhangjianping@caas.cn (J.Z.); yangyongjie@caas.cn (Y.Y.); 2Weed Research Unit, New South Wales Department of Primary Industries, Wagga Wagga, NSW 2650, Australia; hanwen.wu@dpi.nsw.gov.au

**Keywords:** metsulfuron-methyl, ALS enzyme activity, *Polypogon fugax*, glutathione S-transferases, cytochrome P450

## Abstract

*Polypogon fugax* is a common winter weed in China and other Asia countries. We have previously found a *P. fugax* biotype (R) resistant to acetyl co-enzyme A carboxylase (ACCase) herbicides also cannot be effectively controlled by some acetolactate synthase (ALS) herbicides. This study evaluated the level of resistance to four ALS herbicides (metsulfuron-methyl, chlorsulfuron, monosulfuron, pyribambenz isopropyl) in the R biotype and the associated resistance mechanism. The R biotype exhibited moderate level of resistance to metsulfuron-methyl (6.0-fold) compared with the sensitive biotype (S). Sequence analysis of *ALS* gene revealed that two *ALS* genes existed in *P. fugax*. However, no substitution associated with ALS resistance mechanism were found in *ALS* genes between the S and R biotypes. The activity of ALS enzyme isolated from the R biotype was inherently higher and less sensitive to metsulfuron-methyl than the S biotype. Glutathione S-transferases (GST) activity was also less sensitive to metsulfuron-methyl in the R than as the S biotypes. Malathion, a cytochrome P450 (CYP) monooxygenase inhibitor, had much greater synergistic effect with metsulfuron-methyl on the R than as the S plants, reducing the ED_50_ value (herbicide dose to inhibit growth by 50%) of metsulfuron-methyl by 23- and 6-fold, respectively, suggesting that CYP mediated enhanced metabolism might contribute to the resistance to ALS herbicides. These results suggest that metsulfuron-methyl resistance in the R biotype was associated with the up-regulated ALS enzymatic activity and the GST and CYP-mediated enhanced herbicide metabolism.

## 1. Introduction

*Polypogon fugax*, also called Asia minor bluegrass, is a common winter weed species distributed across China and other Asian countries. This annual grass has an extended period of emergence from early November to late December. Its life cycle is highly close to several winter crops, including wheat (*Triticum aestivum* L.), rapeseed (*Brassica napus* L.) and some vegetables [1,2]. *P. fugax* is a competitive weed, especially in moist soil and has become an increasing problem in wheat or rapeseed fields in rotation with rice [3,4,5].

Our previous study showed that a biotype of *P. fugax* collected from Sichuan province in China was highly resistant to ACCase herbicides, including clodinafop-propargyl (1991-fold), fluazifop-p-butyl (364-fold), haloxyfop-R-methyl (269-fold), quizalofop-p-ethyl (157-fold), and fenoxaprop-p-ethyl (8-fold) [6]. ACCase herbicides have been widely used to control a broad range of grass weeds in both dicotyledonous and cereal crops [7,8]. Substitutions in the carboxyltransferase (CT) domain of plastidic *ACCase* genes are associated with target-site resistance to ACCase herbicides [9,10]. Resistance in this *P. fugax* biotype is conferred by an Ile-2041-Asn substitution in ACCase CT domain [6]. The same mutation endows resistance in *Alopecurus myosuroides* [11], *Lolium rigidum* [12] and *Beckmannia syzigachne* [13]. Alternative herbicidal options are therefore needed to control this ACCase-resistant *P. fugax* biotype.

Adoption of annual herbicide rotation and tank mixtures with multiple modes of action is widely promoted for the management of herbicide-resistant weeds [14,15]. However, a diverse chemical control could result in the evolution of multiple-resistant weeds. During the evaluation of alternative herbicide options for *P. fugax* control, we found some ALS-inhibiting herbicides could not effectively control the R biotype (Appendix A). Similarly, biotypes of *Echinochloa crus-galli* var. *formosensis* with resistance to ACCase-inhibiting herbicides were also resistant to ALS-inhibiting herbicides [16]. *L. rigidum* has been reported to have evolved multiple resistance to glyphosate, ACCase and ALS herbicides [12]. Herbicide resistance mechanisms involve target-site resistance (TSR) and non-target-site resistance (NTSR). Since most herbicides are designed based on novel target enzymes or proteins, the mutations or the regulation of target genes are considered as TSR [14]. NTSR results in the decrease of the active herbicide that reaches the target enzyme or binding domain, and usually associates with the reduction of herbicide absorption and translocation [17,18] and the enhancement of herbicide metabolism [19,20].

In this study, we determined the resistance levels of this ACCase resistant biotype of *P. fugax* (R) to four ALS herbicides (metsulfuron-methyl, chlorsulfuron, pyribambenz isopropyl and monosulfuron) and found R biotype was resistant to metsulfuron-methyl. Then we investigated the associated resistance mechanisms to metsulfuron-methyl.

## 2. Results

### 2.1. Whole-Plant Dose Response to ALS Inhibitors

The dose response experiment showed that the R biotype was resistant to metsulfuron-methyl (Figure 1A), with a RI of 6.0, while there were no significant differences between the R and S biotypes in their responses to chlorsulfuron, monosulfuron and pyribambenz isopropyl (Figure 1B–D). This suggests that the R biotype had moderate resistance levels to metsulfuron-methyl. Because the S and R biotypes were collected from the same location, we used the S biotype as the susceptible control to investigate the possible resistance mechanism in the R biotype.

### 2.2. The Analysis of ALS Gene Sequence in S and R Biotypes

Two sequences of *ALS* gene (*PfALS1* and *PfALS2*) were isolated from these *P. fugax* biotypes. The predicted open reading frames (ORF) of these two sequences were identified by ORF finder (https://www.ncbi.nlm.nih.gov/orffinder/ (accessed on 21 June 2020) and compared by BioEdit software. The ORF domain of *PfALS1* had 1929nt (nucleotide) and 642aa (amino acid), and the ORF domain of *PfALS2* had 1923nt and 640aa. The amino acid sequences of *PfALS1* and *PfALS2* have 98% identity (Figure 2A). To our knowledge, this is the first report about the full-length *ALS* gene in *P. fugax* (Genbank accession numbers: MN101598 and MN101599).

To analyze the homology and evolution of *P. fugax ALS* genes, a phylogenetic analysis was performed in *P. fugax* and other plant species (Figure 2B). Basically, the deduced sequence of *PfALS1* and *PfALS2* ORF domain showed at least about 90% identity with *ALS* genes from other grass species, such as *Poa annua* (95%), *L. rigidum* (97%), *Apera spica-venti* (95%), *Alopecurus aequalis* (94%), *A. myosuroides* (94–95%), *Brachypodium distachyon* (93%), *Aegilops tauschii* subsp. *tauschii* (92–93%), *Echinochloa crus-galli* var. *crus-galli* (89–90%), and *Zoysia japonica* (88–89%). Compared with gramineous crops, the sequences of *P. fugax* ALS ORF domains were similar to *Oryza sativa* and *Zea mays* (90%) and *Triticum aestivum* (95%).

### 2.3. ALS Enzyme Activity Assay

To determine whether the resistance is caused by the insensitive ALS enzyme, we measured the ALS activities in the S and R biotypes under metsulfuron-methyl treatment. The ALS enzyme extracted from the R biotype produced twice more acetoin than the S biotype in the absence of herbicide, with1.22 nmol/mg protein in R and 0.54 nmol/mg protein in S plants, respectively. The ALS enzyme activity of the S biotype was more sensitive to metsulfuron-methyl. The content of acetoin was significantly reduced by 77% at 10^4^ nM in the S biotype (0.12 nmol/mg protein), while it was reduced by 32.4% in the R biotype (0.83 nmol/mg protein) (Figure 3). 

### 2.4. GST Activity Assay

The GST (glutathione S-transferase) activity was detected in the untreated and treated S and R plants at 0, 24, 48 and 72 HAT (hours after treatment). GST activity was similar in S and R at 0HAT (44.46 and 36.70 U/mg protein, *p* > 0.05). After the application of metsulfuron-methyl, the activity of GST in S decreased 4–5 fold at 48 and 72 HAT (8.53 and 11.36 U/mg protein, *p* > 0.05). The GST activity also showed a decreasing trend in R, although the differences were not significant (*p* > 0.05) (Figure 4). The results demonstrated that metsulfuron-methyl repressed GST activity at different extent in S and R. 

### 2.5. Effect of Malathion on Metsulfuron-Methyl Resistance

The ED_50_ values of the S and R biotypes with metsulfuron-methyl plus malathion treatment was 1.21 and 2.10 g a.i. ha^−1^ (gram active ingredient per hectare), which were approximately 6-fold and 23-fold lower than that of plants treated with metsulfuron-methyl alone, respectively (Figure 5). It suggested that malathion produced a synergism with metsulfuron-methyl and the effect of malathion plus metsulfuron-methyl on the R biotype was much greater than on the S biotype.

## 3. Discussion

In this study, we detected the resistance level of the R biotype to four ALS inhibitors (metsulfuron-methyl, chlorsulfuron, monosulfuron and pyribambenz isopropyl). The results showed that the R biotype had moderate resistance to metsulfuron-methyl compared with the S biotype, while no significant resistance to other tested ALS inhibitors. This indicated that the R biotype had multiple resistance to ACCase inhibitor herbicides and metsulfuron-methyl. 

To investigate the resistance mechanism of the R biotype to metsulfuron-methyl, the target gene *ALS* gene was first isolated from the S and R biotype. The analysis of sequence showed that two *ALS* genes (*PfALS1* and *PfALS2*) were existed in *P. fugax*. The ORF domains of these two genes shared 98% identity with 10 amino acid substitutions and 2 gaps. Phylogenetic analysis suggested that *ALS* gene from *P. fugax* was relatively close to dryland grass weeds, such as *L. rigidum*, *P. annua* and *A. myosuroides*. Multiple copies of *ALS* gene are present in many weed species, such as *Schoenoplectus juncoides* [21], *Camelina microcarpa* [22], *Echinochloa crus-galli* var. *formosensis* [16] and *Descurainia sophia* [23]. Eight amino acid substitution positions have been reported related to ALS herbicide resistance previously: Ala122, Pro197, Ala205, Asp376, Arg377, Trp574, Ser653, and Gly654 [24]. In this study, no substitution associated with ALS herbicides resistance mechanism was found between the S and R biotypes, suggesting that the mechanism of resistance to metsulfuron-methyl was not caused by the point mutation of the target gene. 

The regulation of target enzyme activity contributes to the resistance mechanism [25]. Xu et al. reported that the sensitivity of the ALS enzyme was reduced in *A. japonicas* in response to mesosulfuron-methyl treatment [26]. Our results showed that ALS enzyme activity of R was inherently overexpressed and less sensitive to metsulfuron-methyl. Therefore, the inherently high enzymatic activity and relative insensitivity of the ALS enzyme to mesosulfuron-methyl might partly contribute to the resistance in the R biotype. 

NTSR contains a part of these pathways which involve in the altered regulation of a range of genes in resistant plants compared to sensitive plants, such as CYP and GST enzyme families [27]. GST catalyzes the conjugation of many diverse substrates to the tripeptide GSH, which is important in herbicide metabolism in many weed and crop species [28]. A resistant *Avena fatua* biotype had higher GST activity to fenoxaprop-P-ethyl than a susceptible biotype [29]. The GST activity of *Myosoton aquaticum* could be induced by tribenuron, with the resistant biotype being more rapidly induced than the sensitive biotype [30]. Our results showed that metsulfuron-methyl did not significantly affect the activity of GST in the R biotype, however, it progressively decreased the activity of GST in the S biotype over time, suggesting GST activity was more sensitive to metsulfuron-methyl in the S than the R biotypes. Thus GST might be involved in the resistance to ALS inhibitors in the R biotype.

CYP catalyzes a wide variety of monooxygenation/hydroxylation reactions. Malathion is a known CYP inhibitor that has been proved to be effective in synergizing ALS herbicides in *Echinochloa colona* [31], *Bromus tectorum* [32] and *L. rigidum* [12]. The addition of malathion to metsulfuron-methyl significantly decreased the plant biomass of the S and R biotypes in comparison with metsulfuron-methyl alone treatment. These results suggest that the addition of malathion led to a greater herbicidal activity of metsulfuron-methyl on the R plants than on as the S plants. There is a large amount of *CYP* genes in plant genomes, and each *CYP* gene participates in various biochemical pathways to produce primary and secondary metabolites [9]. We tested the expression level of three CYP homologous genes in the S and R biotype under metsulfuron-methyl treatment. The results showed that these *CYP* genes were significantly up- or down-regulated in the R biotype, while regulated less actively in the S biotype (Appendix A). This results also supported that CYP-mediated greater antagonistic effect on the R biotype than as the S biotype. The further study should analysis the transcriptome of AS and AR under herbicide treatment to investigate more *CYP* genes regulation network.

In conclusion, both target-site and non-target site resistance mechanisms were involved in metsulfuron-methyl resistance in the R biotype. Although there were no point mutations of the target genes detected in the present study, the up-regulation of ALS enzyme activity was found in the R biotype, which is a target-site resistance mechanism. The GST and CYP-mediated enhanced metabolisms were the non-target site resistance mechanism.

## 4. Materials and Methods

### 4.1. Plant Materials and Growth Condition

Seeds of two *P. fugax* biotypes with distinct resistant status to ACCase herbicides were collected in a wheat field from Qingshen county, Sichuan province, China (29.54° N, 103.48° E). The biotype designated as R had been previously found resistant to ACCase herbicides [6]. The reference, susceptible biotype (S) was collected in a non-cultivated field which was about 2.5 km away from the field where the R biotype was collected. These biotypes were separately cultivated in a greenhouse at the Zhejiang Research Institute of Chemical Industry (30.15° N; 120.03° E), Hangzhou, Zhejiang province, China. Seeds harvested from plants of these biotypes under greenhouse conditions were air dried and stored in paper bags at 4 °C for 3 months to break dormancy, and then stored at room temperature (20 ± 5 °C) until use. 

### 4.2. Whole-Plant Bioassays for Resistance Confirmation

During the initial evaluation, we found the ACCase resistant biotype R was resistant to several ALS inhibitors. To determine the ED_50_ (herbicide dose to inhibit growth by 50%) values of the R and S biotypes to ALS herbicides, seeds of each biotype were sown in 7.5-cm-diameter plastic pots (about 20 seeds per pot) filled with sterile potting medium (mixed vegetable garden soil/cover soil at 4:1 ratio, *v*/*v*) with pH 6.3 and 13.7% organic matter. After emergence, seedlings were thinned to 10 plants per pot and maintained in the greenhouse at approximately 15 °C under natural sunlight. About three weeks after thinning, plants at the 3 leaf-stage were treated with the four ALS herbicides (metsulfuron-methyl, chlorsulfuron, pyribambenz isopropyl and monosulfuron) at various concentrations (Table 1). The aboveground parts of all the plants in each pot were harvested at 4 weeks after treatment (WAT) for fresh biomass determination. Fresh weight was expressed as a percentage of the untreated control to standardize comparisons between populations. Each treatment had three pots as replications, and the experiment was repeated three times.

### 4.3. Isolation and Sequencing of ALS Genes from P. fugax

Leaf tissue was collected from the S and R biotypes at 3–4 leaf-stage and their genomic DNA was extracted using the Plant Genomics DNA Kit (DP320, Tiangen, China) by following the protocol from the manufacturer. All the primers used in this study were shown in Table 2. The partial ALS sequence of *P. fugax* was amplified with primers (ALS569-1246-F/R) designed based on published *ALS* gene sequence of *Alopecurus japonicas* (Genbank: AJ437300). The extracted genomic DNA was used for the pre-amplification with LAD primers and ALS1246RB-0a (ALS569LB-0a) primers, which were designed based on the known partial ALS sequence. The 20-fold diluted pre-amplification products were used as a template in primary TAIL-PCR with primers AC1 and ALS1246RB-1a (or ALS569LB-1a), and then the 20-fold diluted primary TAIL-PCR products were used in the second round of TAIL-PCR with primers AC1 and ALS1246RB-2a (ALS569LB-2a). Because the upstream of *ALS* gene was not completely obtained after the first round TAIL-PCR, the second round TAIL-PCR was processed with ALS110LB (-0a, 1a, 2a) until the full-length of the *P. fugax ALS* gene was obtained. The PCR reaction of each step was prepared as shown in Appendix A, and the thermal conditions were those described by previous report [33]. The second round of TAIL-PCR products were detected with 1% agarose gel, and single fragments were isolated from gel by DNA gel extraction Kit (AP-GX-250, Axygene, Union City, CA, US) and sequenced by Biosune Biotechnology Co., Ltd. (Shanghai, China). 

The *ALS* gene full-length primers (PfALS-F/R) were designed according to the full ALS sequence amplified by TAIL-PCR. Genomic DNA extracted from eight individual plants of each biotype was used as the template for *ALS* gene amplification. The PCR products were detected with 1% agarose gel and extracted. The selected PCR bands were cloned with the pMD19-T vector (6013, Takara, Beijing, China), after which the recombinant plasmids were introduced into competent *Escherichia coli* JM109 (Takara, China) according to the manufacturer’s instructions. At least 15 positive clones of each transformant were sequenced by Biosune Biotechnology Co., Ltd. (Shanghai, China), and the sequence data were aligned and compared by BioEdit software (Version 7.2.5). 

The amino acid sequences of the *PfALS1* and *PfALS2* ORF domain were compared with other grass weeds, gramineous crops, *Arabidopsis* and soybean (*Glycine max*) by BLAST. A phylogenetic analysis was performed by MEGA 5.0 using the neighbor-joining method.

### 4.4. ALS Enzyme Activity Assay

ALS enzyme extraction and activity bioassay were performed as described by Feng et al. with some modifications [34]. Proteins were extracted from fresh leaf tissues of two *P. fugax* biotypes S and R at 3–4 leaf-stage. Five gram of leaf tissues were homogenized with liquid nitrogen and suspended with 20 mL extraction buffer which contained 0.1 M potassium phosphate buffer (pH 7.5), 5 mM Na-pyruvate, 5 mM MgCl_2_, 10 μM flavin adenine dinucleotide (FAD), 1 mM thiamine pyrophosphate (TPP), 10% (*V*/*V*) glycerol. The homogenates were stirred for 30 min on ice, and then filtered by four layers of cheesecloth and centrifuged at 10,000× *g* for 20 min at 4 °C. The supernatant was immediately used for ALS enzyme activity and protein concentration assay. The protein concentration in the enzyme extracts was measured by Bradford method [35]. The ALS activity was determined based on the amount of acetoin formed from acetolactate. Each reaction mixture contained 450 μL enzyme extracts, 50 μL metsulfuron-methyl solution at a series of concentrations and 500 μL reaction buffer (0.1 M potassium phosphate buffer (pH 7.0), 100 mM Na-pyruvate, 5 mM MgCl_2_, 10 μM FAD, 1 mM TPP). A concentration series of metsulfuron-methyl (98%, Aladdin, Shanghai, China) was prepared in acetone at 10^−4^, 10^−2^, 1, 10^2^, 10^4^ nM. The untreated controls received 50 μL acetone. All the reaction mixtures were incubated at 37 °C for 1 h, and then added 20 μL 3 M H_2_SO_4_ and incubated at 60 °C for 15 min to stop the reaction. Next, freshly prepared creatine (5 g/L) and α-naphthol (50 g/L in 2.5 M NaOH) were added and incubated at 60℃ for 15 min. Acetoin concentration was measured by spectrophotometry at 530 nm and the results were adjusted by a standard curve produced by commercial acetoin (99%, Aladdin, Shanghai, China). The background acetoin concentration was subtracted by the blank control that the reaction was stopped with H_2_SO_4_ before the first incubation. ALS activity was expressed as nmol acetion·per mg leaf protein. 

### 4.5. GST Activity Assay

Seedlings of the S and R biotypes at 3-leaf stage were treated with metsulfuron-methyl at 6 g ai ha^−1^ and 0.2 g fresh leaf tissues were collected at 0, 24, 48, and 72 h after treatment (HAT). The 0 HAT samples were used as control. The leaf tissues were homogenized with liquid nitrogen and suspended with 0.8 mL extraction buffer (1× PBS, pH7.4). The homogenates were stirred for 10 min on ice and centrifuged at 10,000× *g* for 5 min at 4 °C. The supernatant was immediately used for GST activity and protein concentration assay. The protein concentration assay was performed as described above. The GST activity assay was performed with the Glutathione S-transferase (GSH-ST) Detection Kit (A004-1-1, Jiancheng, China) by following the protocol from the manufacturer. The activity unit (U/mg leaf protein) of GST was defined as the amount of the enzyme that catalyzes the conversion of μmol of substrate per minute per mg protein. 

### 4.6. Effect of Malathion on Metsulfuron-Methyl Resistance

To determine if malathion, a CYP inhibitor, had effect on metsulfuron-methyl resistance, a whole-plant dose response experiments were carried out as described above. Seedlings of S and AR biotypes at 3-leaf stage were treated with malathion at 1000× *g* a.i. ha^−1^ at 30 min before metsulfuron-methyl application. The metsulfuron-methyl was applied at the same doses as the whole-plant bioassay described above. The aboveground fresh weight was recorded at 4 WAT and the ED_50_ values were calculated. 

### 4.7. Statistical Analysis

All experiments in this study were arranged in a randomized complete block design with three replications for each biotype and herbicide treatment, and the experiment was repeated at least twice. In the whole-plant dose response experiments, a three-parameter logistic equation was fitted to the weed biomass at the various treatments of herbicides combined with their concentrations using the “drc” add-on package in R 3.5.3 (R Core Team, 2019):*Y* = *d*/[1+(*x*/ED_50_)*^b^*]
where *Y* denotes the fresh weight, expressed as percentage of control; *d* is the upper limit of the fresh weight at the dose zero; *b* is the slope at the ED_50_, and ED_50_ is the herbicide dose required for 50% growth reduction. The level of resistance for the *P. fugax* biotypes was determined by the resistance index (RI), which was calculated as the mean value of ED_50_ of the R biotype divided by that of the S biotype.

In the GST activity experiments, analysis of variance was performed via SPSS software, version 13.0 (SPSS, Chicago, IL, USA). Mean comparison was performed using Fisher protected least significant difference (LSD) test, where the overall differences were significant (*p* ≤ 0.05).

## Figures and Tables

**Figure 1 plants-10-01309-f001:**
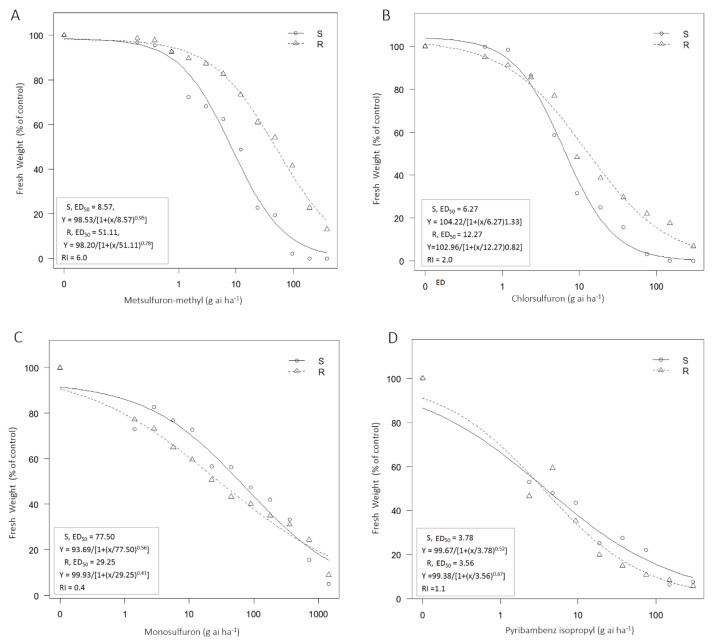
Sensitivities of the S and R *P. fugax* biotypes to different ALS herbicides. (**A**) metsulfuron-methyl; (**B**) chlorsulfuron; (**C**) monosulfuron; (**D**) pyribambenz isopropyl. ED_50_: herbicide dose to inhibit growth by 50%; RI: resistance index.

**Figure 2 plants-10-01309-f002:**
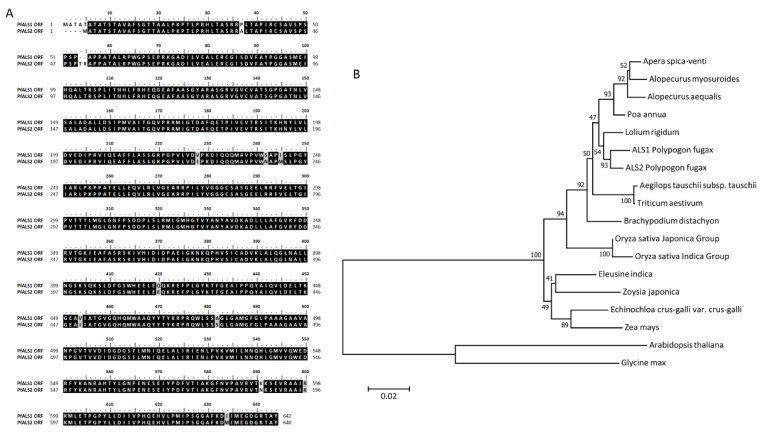
The analysis of PfALS1 and PfALS2. (**A**) The comparison of PfALS1 and PfALS2 ORF domain amino acid sequences; (**B**) The phylogenetic trees of plant ALS proteins created by the neighbour-jointing method with MEGA 5.0. GenBank accession numbers of other ALS proteins are as follows: *Poa annua* (ALE27652.1), *Lolium rigidum* (AIN75605.1), *Apera spica-venti* (AET31400.1), *Alopecurus aequalis* (AFK33197.1), *Alopecurus myosuroides* (CAD24801.2), *Brachypodium distachyon* (XP_003575020.1), *Aegilops tauschii subsp. tauschii*(XP_020168960.1), *Echinochloa crus-galli* var. *crus-galli* (BAQ58050.1), *Zoysia japonica* (BAI44129.1), *Triticum aestivum* (AAO53549.1), *Oryzae sativa* (Japonica Group AAX14282.1 and Indica Group ABF66049.1), *Zea mays* (NP001151761.1), *Arabidopsis thaliana* (CAB62345.1) and *Glycine max* (XP_003543059.1).

**Figure 3 plants-10-01309-f003:**
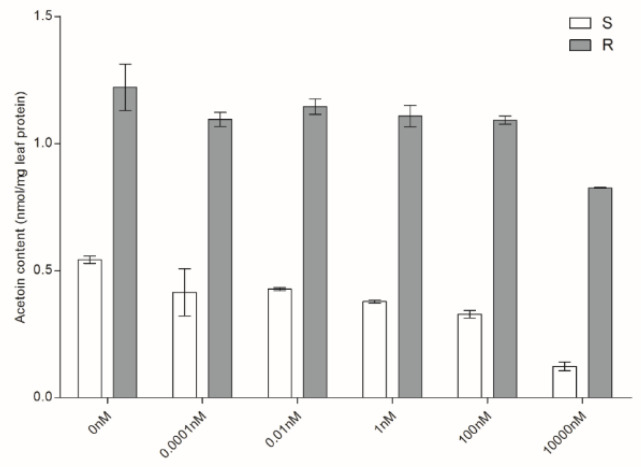
The ALS activity of S and R with various concentrations of metsulfuron-methyl. The *x*-axis represents different doses of metsulfuron-methyl. Bars are mean ± standard error (n = 3).

**Figure 4 plants-10-01309-f004:**
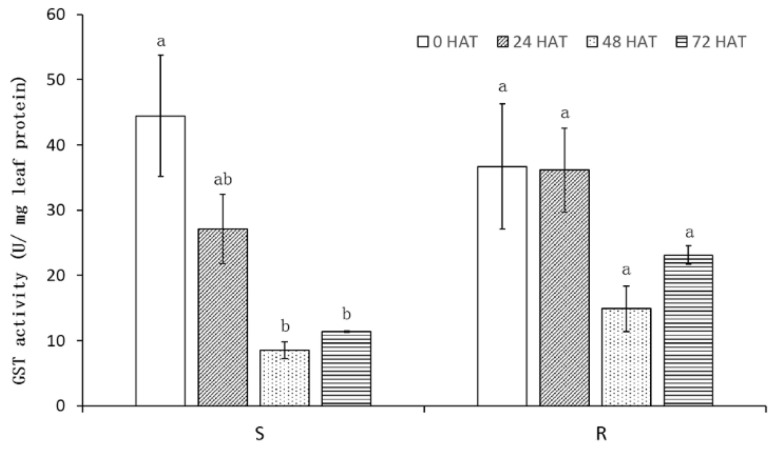
GST activity in S and R at 0, 24, 48, and 72 h after metsulfuron-methyl treated (HAT). Bars are mean ± standard error (n = 3). Values in a column followed by different letters are significantly different based on Fisher’s protected LSD test (*p* ≤ 0.05).

**Figure 5 plants-10-01309-f005:**
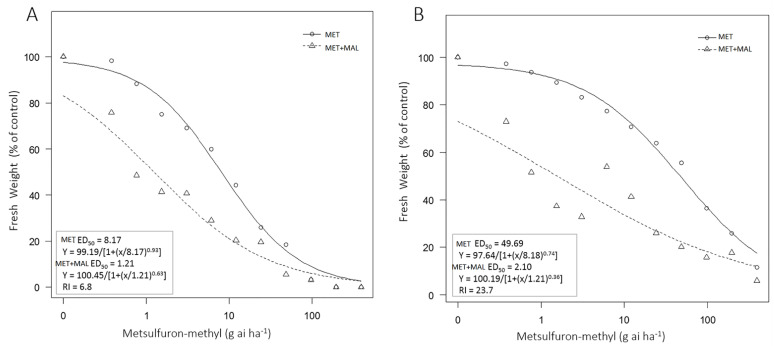
Dose response to metsulfuron-methyl of S and R in the absence or presence of malathion. (**A**) The S biotype; (**B**) The R biotype. MET: only metsulfuron-methyl treatment; MET+MAL: metsulfuron-methyl and malathion treatment. ED_50_: herbicide dose to inhibit growth by 50%; RI: resistance index.

**Table 1 plants-10-01309-t001:** Herbicides used in whole plant dose response experiment.

Herbicide	Manufacturer/Suppliers	Application Rate (g a.i. ha^−1^)
Metsulfuron-methyl	Jiangsu Tianrong Group Co., Ltd., China	0.19, 0.38, 0.75, 1.50, 3.00, 6.00, 12.0, 24.0, 48.0, 96.0, 192.0, 384.0
Chlorsulfuron	Zhejiang Research Institute of Chemical Industry, China	0.59, 1.17, 2.34, 4.69, 9.38, 18.75, 37.50, 75.00, 150.00, 300.00
Pyribambenz isopropyl	Zhejiang Research Institute of Chemical Industry, China	2.3, 4.7, 9.4, 18.8, 37.5, 75.0, 150.0, 300.0
Monosulfuron	Zhejiang Research Institute of Chemical Industry, China	1.41, 2.81, 5.63, 11.25, 22.50, 45, 90, 180, 360, 720, 1440

**Table 2 plants-10-01309-t002:** The primers used in this study.

Primer	Sequence (5′-3′)
ALS569-1246-F	TCACCAAGCACAACTAC
ALS569-1246-R	CTCATGCCACGAACTAA
LAD-1	ACGATGGACTCCAGAGCGGCCGCVNVNNNGGAA
LAD-2	ACGATGGACTCCAGAGCGGCCGCBNBNNNGGTT
LAD-3	ACGATGGACTCCAGAGCGGCCGCVVNVNNNCCAA
LAD-4	ACGATGGACTCCAGAGCGGCCGCBBNBNNNCGGT
AC1	ACGATGGACTCCAGAG
ALS569LB-0a	GATGCAGAGCAGCCACCGCCAACATAAA
ALS569LB-1a	ACGATGGACTCCAGTCCGGCCGTGGCTTGGGCAGGCGGGCAATGTAC
ALS569LB-2a	TGCTGGATGTCTTTGGGGAT
ALS110LB-0a	GGATCCCAGTCAGCTCAACAAATC
ALS110LB-1a	ACGATGGACTCCAGTCCGGCCGAGGAAGAAGGCTTCCTGAATGAC
ALS110LB-2a	GACACCGCGGAGCACCTG
ALS1246RB-0a	TTATGTTGGCGGTGGCTGCTCTGCATC
ALS1246RB-1a	ACGATGGACTCCAGTCCGGCCTTGCATTTGGTGTGCGGTTTGATGAT
ALS1246RB-2a	CAAGAACAAGCAGCCGCATG
PfALS-F	CTCACCCAAACCCTCG
PfALS-R	GCACTTGTCGGTCATGTAG

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
