# Peer review of "Resistance Mechanism to Metsulfuron-Methyl in Polypogon fugax"

_plants, 2021, doi:10.3390/plants10071309_

Round 1

Reviewer 1 Report

Reviewer’s comments

This study deals with the resistance of R and S types of a weed species Polypogon fugax against herbicides. The manuscript was rather well and carefully written but all abbreviations presented in text should be defined when mentioned for the first time. That would help potential readers to understand the story. Results were clear but the presentation of the results could be improved. Explanations in y- and x-axes and in the legends were too small. Furthermore, also in the figures and tables, all abbreviations should be explained carefully. Otherwise it is hard to understand the results. Moreover, some mistakes were found in text (please, see below).

Detailed comments

In title, you have Polypogan fugax but in text Polypogon fugax. Please, correct this mistake.

In Abstract, please, explain what metsulfuron-methyl and ED50 is.

In Table S1, please, define AR and AS.

In the manuscript, you use unit “g ai ha-1”. This is not familiar for me and possibly unclear also for someone else. Could you, please, consider some other way to express this?

In section 2.1, you show results concerning the differences between R and S type after the metsulfuron-methyl treatment. How you can prove this statement? What test was used?

In section 2.2, please, define nt and aa shortly, e.g., within brackets.

In Figure 3, please, add an explanation to x-axis.

In section 2.4, please, define, i.e., give an explanation to GST and HAT. Furthermore, please, add the results from statistical tests to text after each result (p-values, i.e., the statistical significance of the results). In this section you say that: “The GST activity also showed a decreasing trend in R, although the differences were not significant (P ≤ 0.05) (Figure 4).” If you say that the result was not statistically significant, you should write (P > 0.05) instead of (P ≤ 0.05). Same in Figures 4 and S1, where you say that: “Values in a column followed by the same letter are not different according to Fisher’s least significant differences test (P ≤ 0.05).”

In section 2.5, please, define ED50.

In Disucssion, you could again explain abbreviations, especially if they were not mentioned earlier. This sentence is repetition (same was explained already in the previous sentence): “The reduced GYP metabolism of metsulfuron-methyl due to malathion resulted in greater inhibition of both S and R plants, with the R plants being suppressed more than as the S plants.” Why CYP gene analyses are not presented already in Results, although they clearly belong to that section?

In section 4.1, should the reference Tang et al. 2014 be expressed as a number as is done for the other references?

In Methods, why fresh weight of plants was used in analyses? I think that dry weight would have been more reliable. In Methods, you give an impression that ALL plants were treated using ALL herbicides. Thus, please, indicate clearly that different plant groups were treated using different herbicides. In Table 2, please, try to put one sequence to one row. Why untreated control plants received 50 µl acetone?

Author Response

Response to Reviewer 1

In section 2.1, you show results concerning the differences between R and S type after the metsulfuron-methyl treatment. How you can prove this statement? What test was used?

Response: We used the whole-plant dose experiment to determine the ED50 values of R and S biotypes to metsulfuron-methyl. According to Figure 1, the ED50 values of R and S type to metsulfuron-methyl 51.11 and 8.57. The resistance index was 6-fold, suggesting that R type is resistant to metsulfuron-methyl. However, the resistance index of chlorsulfuron, monosulfuron and pyribambenz isopropyl were 2.0, 0.4 and 1.1, suggesting that R type is not resistant to these three herbicides.

Why CYP gene analyses are not presented already in Results, although they clearly belong to that section?

Response: The antagonistic effect of malathion on metsulfuron-methyl suggested that CYP mediated metabolism contribute to the resistance mechanism. The different expression of CYP genes in R and S biotypes also supported this results. But I only tested three CYP genes expression level, it could not reflect the global regulation of CYP family under metsulfuron-methyl treatment. Further study like transcriptome analysis should be conducted to investigate more related CYP genes.

In Methods, why fresh weight of plants was used in analyses? I think that dry weight would have been more reliable.

Response: The weight of P.fugax plant is relatively light. So the fresh weight is more accurate than dry weight. Many studies also used fresh weight to calculated ED50 values.

Why untreated control plants received 50 µl acetone?

Response: Because metsulfuron-methyl was dissolved in acetone. The control plants treated with 50 µl acetone to prove that acetone had no effect on ALS activity.

Besides these, I also made some modifications followed your suggestions. Please check the revised manuscript.

Reviewer 2 Report

A very good article. Congratulations to the authors.

Author Response

Response to reviewer 2

The title have been corrected.

The last sentence of section 2.2 have been deleted.

In discussion, “Descurainia Sophia” have been corrected as “Descurainia sophia”.

Reviewer 3 Report

The main question addressed by the research is the determination the resistance levels of this ACCase resistant biotype of P. fugax (R) to four ALS herbicides (metsulfuron-methyl, chlorsulfuron, pyribambenz isopropyl and monosulfuron), and to investigate the associated resistance mechanisms.

The study subject is relevant and interesting, because it discussed to a global problem relating the weed control and search of alternative herbicidal options. This research also contributes to the plant biochemistry by explaining the mechanism of action of four commercial herbicidal substances and resistance of Asia minor bluegrass, which is not yet fully understood.

Modern methods and techniques are applied to identify a ALS gene sequence, to assay the enzymatic activity and for statistical evaluation of the results, like PCR, spectrophotometry, “drc” add-on package in R 3.5.3 (R Core Team, 2019), etc.

The paper is well written, the text is clear and easy to read. The conclusions consistent with the evidence and arguments presented, and they address the main question posed.

I have some questions and suggestions, e.g.:

Title: There is defined the achieved goal: „Resistance Mechanism to Metsulfuron-methyl in Polypogan fugax“. But in the paragraph 1. “Introduction” the formulation of the purpose of the study is: “In this study, we aimed to determine the resistance levels of this ACCase resistant biotype of P. fugax (R) to four ALS herbicides (metsulfuron-methyl, chlorsulfuron, pyri-bambenz isopropyl and monosulfuron), and to investigate the associated resistance mech-anisms.”

            There, I find an unconformity. Please, rewrite the aim of the study in the “Introduction”.

Paragraph 4.3:

Leaf tissue was collected from the S and R biotypes at 3-4 leaf-stage and their genomic DNA was extracted using the Plant Genomics DNA Kit (Tiangen, China) by following the protocol from the manufacturer.”

Is this protocol a standard method, or were it validated? Please, provide a reference source.

Paragraph 4.5:

The GST activity assay was performed with the Glutathione S-transferase (GSH-ST) Detection Kit (Jiancheng, China) by following the protocol from the manufacturer.

Please, provide a reference source.

Paragraph “References”: There are included publications from a large time period (from 1994 to 2018), and only less than 9% are from the last 5 years.

Can the author find some more publications from the last few years to show this work even more up-to-date.

Author Response

Response to reviewer 3

Is this protocol a standard method, or were it validated? Please, provide a reference source.

Response: The protocols of plant genomics extraction and GST detection are standard method. I added the product number of each Kit. The protocols could be downloaded on the website.

Besides these, I also made some modifications followed your suggestions. Please check the revised manuscript.